# **Differentiating Diffuse Aurora Based on Phenomenology**

Eric Grono<sup>1</sup> and Eric Donovan<sup>1</sup> <sup>1</sup>University of Calgary, Calgary, Alberta, Canada **Correspondence:** Eric Grono (emgrono@ucalgary.ca)

# Abstract.

5

There is mounting evidence which suggests that pulsating auroral patches often move with convection. This study is an initial step at identifying the differences between patches that move with convection and those that do not. While many properties of pulsating patches vary, here we outline criteria for separating pulsating auroral patches into three categories based on two properties: their structural stability and the spatial extent of their pulsations. Patchy aurora is characterized by stable structures whose pulsations are limited to small regions. Patchy pulsating aurora consists of stable patches whose pulsations are far less subtle and occur throughout much of their area. Amorphous pulsating auroral structures are unstable – very rapidly evolving – and can pulsate over their entire area. The speed with which amorphous pulsating aurora evolves makes their motion difficult to ascertain and seems unrelated to the  $E \times B$  drifting of cold, equatorial plasma.

# 10 1 Introduction

The immediate cause of aurora is the precipitation of electrons and protons into the atmosphere. These charged particles collide with atmospheric particles and energize them, whereupon they may spontaneously emit a photon. One basis upon which aurora can be categorized is the mechanism that drives electrons and protons to enter the atmosphere and produce aurora. *Discrete aurora* describes the loss of magnetically bounce-trapped particles into the atmosphere due to electric fields

15 parallel to the magnetospheric magnetic field. *Diffuse aurora* is created through particle loss driven by the pitch-angle scattering of magnetically bounce-trapped electrons and protons.

Pulsating aurora is a type of diffuse aurora which is characterized by quasi-periodic transitions between bright and dim states typically recurring on the order of 10s whose precipitating electrons have energies ranging from a few keV to hundreds of keV (Johnstone, 1978). This type of aurora is widely believed to arise from cyclotron resonance between plasma waves and

20 equatorial electrons (e.g. Coroniti and Kennel, 1970; Davidson, 1990; Miyoshi et al., 2010), however, the details of this process are currently an open area of research. Nishimura et al. (2010, 2011) presented simultaneous observations of auroral luminosity from all-sky imager (ASI) data and chorus wave amplitude measurements from spacecraft that demonstrated a nearly one-toone correspondence. This work confirmed the long-held belief that electrons with energies on the order of 100 eV–10 keV are able to resonate with electron cyclotron harmonic (ECH) and whistler mode chorus waves. Individual pulsating auroral structures can vary in shape, size, altitude, spatial stability, temporal modulation, lifespan, and velocity. An extensive survey of pulsating aurora by Royrvik and Davis (1977) established three general morphologies: east-west aligned arcs, quasi-linear elements, and patches. Pulsating auroral patches are the most common form of this aurora.

Pulsating aurora is most commonly seen in the morning sector auroral oval and persists for 1.5 h on average (Jones et al., 2011) but has been observed to last upwards of 15 h (Jones et al., 2013). The lifetime and size of pulsating auroral forms are known to vary substantially. Lifetimes can span a single pulse that lasts for only a few seconds or persist for tens of minutes. The pulsating auroral arc is on the order of 1–10 km wide and 1000 km long; the quasi-linear form has width on the same order as the arc but length on the order of 100 km. Pulsating patches tend to be 10–200 km across (Royrvik and Davis, 1977) and

have a constantly evolving shape (Shiokawa et al., 2010).

- Pulsating auroral patches appear to be controlled by structures in the near-equatorial cold plasma (Rae, K., 2014) whose motion is almost entirely determined by  $E \times B$  drifting. Consequently, these patches appear to move with ionospheric convection and could be used to create automatically generated two-dimensional maps of convection (Yang et al., 2015; Grono et al., 2017).
- At least two types of pulsating auroral patches have been discussed in the past; Royrvik and Davis (1977) subcategorized patches based on a property they referred to as 'stability'. The stability of a pulsating auroral form is assessed based on whether the area of the pulsating aurora changes while switching between bright and dim states. Royrvik and Davis (1977) uses the term *stable* to describe structures whose area remains constant while switching between states (Figure 1a), and *streaming* to describe those whose area changes (Figure 1b). This property is dependent on the cadence of the camera and it is likely that all patches exhibit some amount of streaming while pulsating.
- Grono et al. (2017) noted that all structures that would be considered to be a pulsating auroral patch did not move with convection. They identified a type of patch whose motion was more dynamic and whose morphology was visually distinct from those that followed convection. This study will describe the distinction between these structures and outline a scheme for separating pulsating auroral patches into at least three categories based on the spatial extent of their pulsations as well as the stability of their shape.

# 25 2 Instrumentation and Data

The pulsating aurora events featured in this paper were imaged by the Time History of Events and Macroscale Interactions during Substorms (THEMIS) all-sky imager (ASI) array (Donovan et al., 2006; Mende et al., 2008), the ground-based component of the NASA mission (Angelopoulos, 2008) focused on the study of aurora and substorms. The network currently consists of 21 ASIs stationed across northern North America which capture "white light" images of the aurora on a 256x256 pixel CCD

30 with a three second cadence. This network has been operating for over ten years and has amassed tens of millions of images. As demonstrated by Grono et al. (2017), it can be extrapolated from Jones et al. (2011) that on the order of 10% of THEMIS ASI images contain pulsating aurora.