# Peer review of "Differentiating Diffuse Aurora Based on Phenomenology"

_Annales Geophysicae, 2018_

## Referee Comment (RC1) · Anonymous Referee #1 · 16 Mar 2018

This manuscript reports results of a pulsating aurora study, where THEMIS ASI data are used to classify pulsating aurora events into three different subgroups based on the patch size and stability. The third pulsating aurora class is found to exhibit motion which does not follow the ionospheric convection drift. Pulsating aurora has been studied a lot recently but little is done about the structural evolution or differences within that type of aurora. That makes this study very welcome and well worth publishing. The manuscript is compact and flows well. Thus, I only have some minor comments to consider before publication.

Clarification requests:

- p.1 line 17: "quasi-periodic transitions between bright and sim states" is interesting. This is often referred to as on/off states, which is also later used in the manuscript. Based on the provided data, the authors seem right in that it is not necessarily on/off but rather fluctuations in luminosity. Maybe this is worth commenting in the paper as well?
- p.1 line 18: The precipitation energies associated with pulsating aurora have been observed to reach some hundreds of keV (e.g. Miyoshi et al. 2015). An order of magnitude increase for the upper end of the energy would be welcome here.
- p.3 line 1: There are plenty of keograms in the paper by Eather et al. (1976) but pulsating aurora is not mentioned, or how it might look like in a keogram is not discussed, so this may not be a proper reference. Later on the pulsating aurora as seen in keograms is given references to Jones et al., Partamies et al and Yang et al. Any of those papers would make a better reference, since they all show and describe how pulsating aurora looks like in keogram data.
- Categories of patches: Can amorphous pulsating aurora structures be called patches in the same way and meaning the word patch is used for the other 2 categories? It is explicitly said that these are the ones which are difficult to track, while patches of PA/PPA are trackable. This leads to another question: What is the role of the identification/tracking challenge in category 3 when it comes to the conclusion that these features do not drift along the ionospheric convection? If they cannot be tracked how reliable is their drift speed estimate?
- Discussion: The authors conclude that the types of pulsating aurora reported in earlier papers have been PA/PPA. Does this mean that a detailed investigation of previously published keograms/images has been carried out to draw this conclusion? If yes, it is an important process to be described in the manuscript.
• Figures: The first reference to Figures 3 and 4 comes on page 4, before Figure 2 has been introduced. Instead of referring to figures 3 and 4 one could give a general event selection description: How many days/nights of data? Which stations? How those were selected? What I also wonder in this context is whether anything could be said about the MLT or latitude distribution of the different event types?

Text and typos

- p.1 line 13: "mechanisms that motivates electrons" would require a more suitable verb
- p.2 line 14: "patches" instead of "patch"
- p.2 line 29: "panchromatic white light" sounds overdoing the statements, since the two terms mean about the same.
- p.4 line 7: Why is the supplementary material cited as 2017 if it is related to this manuscript?
- p.9 line 32: "there" instead of "their"

| AN | GE | OD |
|----|----|----|
|----|----|----|

---

## Referee Comment (RC2) · Anonymous Referee #2 · 11 Apr 2018

General Comments

The manuscript begins to classify pulsating aurora into three types: a very difficult and much-needed systematic approach. The writing flows well and the results are clear, although I'm not sure they bring very much to the state of our understanding. I hoped for more quantitative conclusions, such as thresholds for how many subsequent 3-s images a patch must maintain its shape to be considered patchy pulsating aurora. I would also like to see more statistical analysis, at least of the events you have already analyzed for this paper. How often was patchy aurora seen compared to patchy pulsating and amorphous pulsating? These details would vastly improve the usefulness of the results.

Otherwise, the figures and data quality looks very good, and the presentation is clear

and concise. At this point, I would say the manuscript may have potential after additional work and resubmission.

Specific Comments

All page number (P) and line numbers (L) refer to the edited paper with tracked changes as shown in the reply to Reviewer Comment 1 (AC1 Supplement).

P1, L24-26: "Higher energy electrons on the order of 1–100 keV are thought to also be affected by electromagnetic ion cyclotron (EMIC) waves (e.g. Ni et al., 2016)." This is not true. Relativistic electrons (typically multi-MeV) may be resonant with EMIC waves, but certainly nothing in the pulsating auroral energy regime.

P4, L5-6: Does the identification requirement include those patches nearest to zenith? If so, please state that as a factor in being "most dominant".

P4, L33-34: Please specify if these examples are subsequent individual frames (indicating a 3 second resolution) or if frames are skipped. From the total time duration, it seems that these are subsequent frames, but just want to have that be clear.

P5, L4: So patchy aurora is not fluctuating periodically in brightness. Is my understanding correct?

Technical Corrections

Minor typos/grammar:

P1, L19: 'aurora' twice

P2, L18, 'dependent' should be 'dependent'

P6, L29: Change 'who' to 'that' or 'which'

P9, L14: 'patch' should be 'patches'

---

## Author Comment (AC2) · 13 Apr 2018

Attached is the combined response to the comments of referees 1 and 2. The marked-up manuscript contained within are the differences between the latest version and the original submission. Our responses to both referees are included.

Please also note the supplement to this comment:
https://www.ann-geophys-discuss.net/angeo-2018-21/angeo-2018-21-AC2-supplement.pdf

---

## Author Response (AR1)

**General Comments:**

The manuscript begins to classify pulsating aurora into three types: a very difficult and much-needed systematic approach. The writing flows well and the results are clear, although I'm not sure they bring very much to the state of our understanding. I hoped for more quantitative conclusions, such as thresholds for how many subsequent 3-s images a patch must maintain its shape to be considered patchy pulsating aurora.

*Generally, very few images are necessary to make the distinction. Based on Figure 2, it is possible to conclude only two images were necessary in this instance, but to be certain it is better to consider more. Sometimes events can be ambiguous since there seems to be gradation between types.*
*Clarifying information was added to paragraphs 4-6 of Categories of Patches.*
*APA – patches often exist for only one frame.*
*PA & PPA – tens of minutes*

I would also like to see more statistical analysis, at least of the events you have already analyzed for this paper. How often was patchy aurora seen compared to patchy pulsating and amorphous pulsating? These details would vastly improve the usefulness of the results. Otherwise, the figures and data quality looks very good, and the presentation is clear and concise. At this point, I would say the manuscript may have potential after additional work and resubmission.

*The focus of this work is to present the phenomenological argument for subcategorizing pulsating aurora. Moving forward we fully intend to quantitatively investigate pulsating aurora by doing a comprehensive survey of THEMIS ASI data for several imagers. This set of pulsating aurora events was identified through a quick survey of the ASI data looking for 32 ~3 hour events in 2011, and since the data was searched chronologically, they may be the first such events of the year for these imagers. A much more thorough survey of years of data is necessary to derive reliable statistical results. The approximate number of hours each type appeared within Figures 3 and 4 is included in paragraphs 4-6 of Categories of Patches.*

**Specific Comments:**

All page number (P) and line numbers (L) refer to the edited paper with tracked changes as shown in the reply to Reviewer Comment 1 (AC1 Supplement).

P1, L24-26: "Higher energy electrons on the order of 1–100 keV are thought to also be affected by electromagnetic ion cyclotron (EMIC) waves (e.g. Ni et al., 2016)." This is not true. Relativistic electrons (typically multi-MeV) may be resonant with EMIC waves, but certainly nothing in the pulsating auroral energy regime.

*We agree and thank you for bringing this to our attention.*

P4, L5-6: Does the identification requirement include those patches nearest to zenith? If so, please state that as a factor in being "most dominant".

*Rephrased the sentence: "Regions of each figure have been classified based on the type of pulsating auroral patch that appears most prevalent within the ASI field of view during a given period."*

P4, L33-34: Please specify if these examples are subsequent individual frames (indicating a 3 second resolution) or if frames are skipped. From the total time duration, it seems that these are subsequent frames, but just want to have that be clear.

*Fixed. I have specified the images are sequential at the location you specified as well as in the figure caption.*

P5, L4: So patchy aurora is not fluctuating periodically in brightness. Is my understanding correct?

*Patchy aurora events are not necessarily entirely without pulsations, but they tend to be constrained to small regions along the edge of a larger non-pulsating patch, or near one.*

**Technical Corrections:**
Minor typos/grammar

P1, L19: 'aurora' twice
*Fixed.*

P2, L18, 'dependent' should be 'dependent'
*Fixed.*

P6, L29: Change 'who' to 'that' or 'which'
*Changed to "which".*

P9, L14: 'patch' should be 'patches'
*Fixed.*

This manuscript reports results of a pulsating aurora study, where THEMIS ASI data are used to classify pulsating aurora events into three different subgroups based on the patch size and stability. The third pulsating aurora class is found to exhibit motion which does not follow the ionospheric convection drift. Pulsating aurora has been studied a lot recently but little is done about the structural evolution or differences within that type of aurora. That makes this study very welcome and well worth publishing. The manuscript is compact and flows well. Thus, I only have some minor comments to consider before publication.

Clarification requests:

• p.1 line 17: "quasi-periodic transitions between bright and sim states" is interesting.  This is often referred to as on/off states, which is also later used in the manuscript. Based on the provided data, the authors seem right in that it is not necessarily on/off but rather fluctuations in luminosity. Maybe this is worth commenting in the paper as well?

*I changed references to the states being "on" and "off" to "bright" and "dim" for consistency.  I have added a paragraph about this at: Discussion Paragraph 5.*

• p.1 line 18: The precipitation energies associated with pulsating aurora have been observed to reach some hundreds of keV (e.g. Miyoshi et al. 2015). An order of magnitude increase for the upper end of the energy would be welcome here.

*Agreed, changed.*

• p.3 line 1: There are plenty of keograms in the paper by Eather et al. (1976) but pulsating aurora is not mentioned, or how it might look like in a keogram is not discussed, so this may not be a proper reference. Later on the pulsating aurora as seen in keograms is given references to Jones et al., Partamies et al and Yang et al. Any of those papers would make a better reference, since they all show and describe how pulsating aurora looks like in keogram data.

*This reference was intended to point toward the origin of the keogram figure, and not that pulsating auroral patches are easily identified within them, but I see your point.  I moved the Eather reference to the next sentence and replaced it with the Jones et al., Partamies et al., and Yang et al. references.*

• Categories of patches: Can amorphous pulsating aurora structures be called patches in the same way and meaning the word patch is used for the other 2 categories? It is explicitly said that these are the ones which are difficult to track, while patches of PA/PPA are trackable. This leads to another question: What is the role of the identification/tracking challenge in category 3 when it comes to the conclusion that these features do not drift along the ionospheric convection? If they cannot be tracked how reliable is their drift speed estimate?

*I would not call it a patch in the same sense as the two patchy auroras – and I try to emphasize this by the way they are named: patchy, patchy pulsating, and amorphous pulsating auroras.  However, I describe all three types as categories of "pulsating auroral patch", because they all fit the general description of patches within all-sky cameras: that they're irregular 'patchy' structures and have quasi-periodic pulsations.*

*The drift speed estimate of amorphous pulsating aurora is not accurate (but possibly not without value) since they do not seem to drift. The separation criteria gets you the good stuff (stable patch tracks) from the junk (amorphous patch tracks).*

*I have updated Discussion paragraph 1 to clarify this.*

• Discussion: The authors conclude that the types of pulsating aurora reported in earlier papers have been PA/PPA. Does this mean that a detailed investigation of previously published keograms/images has been carried out to draw this conclusion? If yes, it is an important process to be described in the manuscript.

*A detailed investigation was not necessary, it is possible to determine this by looking at the figures in those papers, but I did confirm by looking at the full images. I have added videos of these events to the supplement; these combined with the original figures are sufficient for classification. I have updated Discussion paragraph 2 to clarify this.*

• Figures: The first reference to Figures 3 and 4 comes on page 4, before Figure 2 has been introduced. Instead of referring to figures 3 and 4 one could give a general event selection description: How many days/nights of data? Which stations? How those were selected? What I also wonder in this context is whether anything could be said about the MLT or latitude distribution of the different event types?

*Instrumentation and Data paragraph 3 has been updated to better describe the keograms without referring to them as Figures 3 and 4 and explained the selection criteria.*

*We will be pursuing MLT and latitude distributions, but not for this paper.*

Text and typos

• p.1 line 13: "mechanisms that motivates electrons" would require a more suitable verb

*Changed.*

• p.2 line 14: "patches" instead of "patch"

*Changed.*

• p.2 line 29: "panchromatic white light" sounds overdoing the statements, since the two terms mean about the same.

*Changed.*

• p.4 line 7: Why is the supplementary material cited as 2017 if it is related to this manuscript?

*This paper was initially submitted to GRL in late December 2017 where they declined to send it out for review after questioning the need for rapid publication. I published the supplement in December to coincide with its initial submission.*

• p.9 line 32: "there" instead of "their"

*Changed.*

**Differentiating Diffuse Aurora Based on Phenomenology**

Eric Grono[1] and Eric Donovan[1]

[1]University of Calgary, Calgary, Alberta, Canada

**Correspondence:** Eric Grono (emgrono@ucalgary.ca)

**Abstract.**

[revised manuscript text omitted]

(a) Stable patch
WHIT 20051228          13:26:39 -13:26:54 UT

(b) Streaming patch
ATHA 20110302          07:34:00 - 07:34:15 UT

**Figure 1.** (a) An example of a stable pulsating auroral patch as described by Royrvik and Davis (1977). The images of this patch were captured  during the minute of 13:26 UT on 28 December 2005 from Whitehorse THEMIS ASI. (b) An example of a streaming pulsating auroral patch as described by Royrvik and Davis (1977). The images of this patch were captured  during the minute of 07:34 UT on 02 March 2011 from Athabasca THEMIS ASI. Streaming describes how the area of a patch can grow or shrink while switching between bright and dim states.

Pulsating auroral patches are easily identified in keograms (e.g. Jones et al., 2013; Partamies et al., 2017; Yang et al., . These figures illustrate the spatial and temporal evolution of a north-south aligned column of pixels and are created by isolating a column of pixels in a sequence of images and arranging them in chronological order (Eather et al., 1976). Ewograms, or east-west-keograms, are a variation of this figure made by extracting an east-west aligned row of pixels from a sequence of images. Within keograms and ewograms pulsating aurora can appear as structures featuring quasi-periodic vertical striations which arise due to the aurora alternating between dim and bright states.

 This study contains 32 keograms of pulsating aurora events  which appeared across 27 nights between 1 January 2011 and 1 March 2011 and were imaged by ASIs stationed at Athabasca, Fort Simpson, Fort Smith, Fort Yukon, Gillam, and Sanikiluaq. These events were chosen because they were the first 32 pulsating aurora events in 2011 which persisted for at least two hours and were clearly identifiable as pulsating aurora during a quick survey of keograms. Regions of each figure have been classified based on the type of pulsating auroral patch that  appears most prevalent within the ASI field of view during a given period. These classifications are based on the type of structures that are visible within the keograms as well as the full images and do not imply that other types of patch or aurora are not simultaneously present within the  field of view. The  boundaries of each region are placed imprecisely as their primary purpose is to aid the reader in recognizing the distinct signatures of each type of patch as well as their relative frequency. Videos containing the full all-sky images that the keograms are constructed from are available as a supplement to this paper (Grono, 2017).

~~Sixteen pulsating aurora events where regions of amorphous pulsating aurora, patchy pulsating aurora, and patchy aurora are approximately outlined in red, blue, and yellow, respectively. The outlined regions designate only the most prominent types in the keogram and do not imply that no other types are simultaneously present within the ASI field of view. Videos of the associated ASI data are included as a supplement to this study (Grono, 2017).~~

~~Sixteen pulsating aurora events where regions of amorphous pulsating aurora, patchy pulsating aurora, and patchy aurora are approximately outlined in red, blue, and yellow, respectively. The outlined regions designate only the most prominent types in the keogram and do not imply that no other types are simultaneously present within the ASI field of view. Videos of the associated ASI data are included as a supplement to this study (Grono, 2017).~~

**3    Categories of Patches**

Pulsating auroral patches differ from each other in terms of shape, size, brightness, pulsation frequency and coherence, among other factors. It can be challenging to categorize patches since their characteristics often exhibit gradation between types. This study will establish criteria to classify pulsating auroral patches based on the spatial extent of their pulsations as well as the stability of their shape – not to be confused with the definition of stability utilized by Royrvik and Davis (1977).

The defining characteristic that separates stable and unstable pulsating patches is the ability to identify a singular patch across an extended period of time. To describe a patch as stable in this context means that its shape and intensity evolve slowly enough that it can be easily identified across many THEMIS all-sky images. The shape and intensity of an unstable patch evolve so rapidly that it is challenging – if not impossible – to uniquely identify a patch across multiple images. In addition, the region of a patch that pulsates can vary in spatial extent; pulsations can cover the entire area of a patch or be limited to a smaller region. These two properties, the stability and the spatial extent of pulsations, can be used to separate patches into three categories: *patchy aurora*, *patchy pulsating aurora*, and *amorphous pulsating aurora*.

[Figure]

**Figure 2.**  Series of images containing (a) patchy aurora, (b) patchy pulsating aurora, and (c) amorphous pulsating aurora as well as difference images highlighting the intensity modulations between sequential frames. Red and blue pixels indicate brightening and dimming, respectively.

Figures 2a, 2b, and 2c include examples of the three types to illustrate the differences in stability and pulsation extent between these types of pulsating auroral patch. Each example consists of a  series of all-sky images and  corresponding difference images highlighting the intensity modulations between sequential frames. The difference images are created by subtracting an image from its predecessor and then shifting and byte-scaling the data values until 0 is represented

by white pixels on a blue-white-red colour scale. Red pixels are brighter in the second image than the first and the opposite is true of blue pixels. The images composing Figure 2 are each 100 pixels square and are off-centre from zenith.

Patchy aurora (Figure 2a) consists of stable patches which have a slowly evolving structure that allows individual patches to be easily followed  through their transit of the ASI field of view for tens of minutes. The pulsations visible within a patchy aurora event are often limited to small regions along the edge of the patch or constrained to smaller structures separate from the main body of the patch. In Figure 2a the intensity of the large patch on the left side of the images remains quite constant while the smaller structure on the right exhibits modulation. These pulsations are not referenced in the name of this type of patch since the larger non-pulsating structures are the defining characteristic of this form when considered in comparison to the other two types. Figures 3 and 4 contain approximately 10 hours of patchy aurora.

Patchy pulsating aurora (Figure 2b) also consists of stable patches which are easily trackable; however, in contrast to patchy aurora, these patches pulsate over a larger region – often their entire area. Patchy pulsating aurora can be  easily identified within sequences of all-sky images  for tens of minutes, similar to patchy aurora, but individual structures may not necessarily be visible in each image due to their larger pulsating area. Furthermore, the shape of patches that stream between their  bright and dim states can differ between images and consequently increase the difficulty of identification. Figures 2a and 2b highlight the different pulsating areas of patchy aurora and patchy pulsating aurora; both sequences of images contain persistent, easily recognizable structures but 2b pulsates over a significantly larger region. The approximately 19 hours of patchy pulsating aurora in Figures 3 and 4 is nearly double the amount of patchy aurora which is visible.

In contrast to the two patchy auroras, amorphous pulsating aurora (Figure 2c) consists of transient, irregularly shaped structures which are often impossible to identify and track through sequences of images. Amorphous pulsating aurora generally pulsates over its entire area, exhibits noticeable streaming, and has a rapidly evolving shape demonstrated in Figure 2c.  In fact, it is common that amorphous patches are only uniquely identifiable within a single THEMIS all-sky image. Across the 32 pulsating aurora events in Figures 3 and 4, approximately 67 hours of amorphous pulsating aurora are visible, making these structures the most common type of pulsating patch. Furthermore, amorphous pulsating aurora appears during every pulsating aurora event in Figures 3 and 4, and while it is possible that an event may exclusively feature amorphous pulsating aurora, this is not true of the two patchy auroras. Assuming visibility allows sufficient observations, amorphous pulsating aurora appears to always precede the onset of the other types and may also be seen later.

**4   Discussion**

Yang et al. (2015), the inspiration for Grono et al. (2017), compared five pulsating auroral patches to SuperDARN measurements and demonstrated that patches move with convection. Reconsidering these patches with the categorization scheme described above reveals them to be patchy aurora and patchy pulsating aurora, from which the conclusion can be drawn that the patchy auroras move with convection. In contrast to these, the motion of amorphous pulsating aurora is far more dynamic and

[Figure]

**Figure 3.** Sixteen pulsating aurora events where regions of amorphous pulsating aurora, patchy pulsating aurora, and patchy aurora are approximately outlined in red, blue, and yellow, respectively. The outlined regions designate only the most prominent types in the keogram and do not imply that no other types are simultaneously present within the ASI field of view. Videos of the associated ASI data are included as a supplement to this study (Grono, 2017).

challenging to ascertain , a fact reflected by — amorphous patches do not appear to drift but instead may be transient structures

[Figure]

Figure 4. Sixteen pulsating aurora events where regions of amorphous pulsating aurora, patchy pulsating aurora, and patchy aurora are approximately outlined in red, blue, and yellow, respectively. The outlined regions designate only the most prominent types in the keogram and do not imply that no other types are simultaneously present within the ASI field of view. Videos of the associated ASI data are included as a supplement to this study (Grono, 2017).

which appear and disappear chaotically. Due to the difficulty of  uniquely identifying an amorphous patch throughout a sequences of images it is possible that they exist for only a single pulsation.

This differentiation of pulsating patches was initially noted by Grono et al. (2017) which, while developing an automated method of tracking patches, recognized that all structures that could be considered pulsating auroral patches did not move in a manner consistent with convection. Although these structures were not identified as such by Grono et al. (2017), similar to Yang et al. (2015) the patches that they studied were patchy aurora and patchy pulsating aurora and those that were noted to not follow convection were amorphous pulsating aurora. The original figures from these papers as well as the full all-sky images for these events – the latter of which are included in Grono (2017) as videos – provide sufficient information to classify these events.

The tracking algorithm of Grono et al. (2017) can be used to automatically separate the structures that move with convection from those that do not within series of images containing pulsating auroral patches. This technique functions by applying a wavelet filter to series of images to make auroral structures Gaussian-like and then following them between images to produce 'tracks' under the assumption they behave like non-interacting Brownian particles. The tracks produced by this algorithm can then be used to classify structures within the images. The threshold that has been found empirically to separate the patchy auroras from other tracks is based on two criteria: track length and the standard deviation of patch velocity in the magnetic local time (MLT) frame of reference. A minimum track length of 60 images – corresponding to a minimum patch lifetime of 3 minutes at the 3 second cadence of THEMIS ASIs – combined with a maximum standard deviation of 1000 has been found to be an effective threshold for identifying patchy auroras within a pulsating aurora event.

In its current state, separating patches with this technique requires pulsating aurora events to be manually identified before being processed by the tracking algorithm. It is computationally wasteful to track ASI data that does not feature pulsating aurora, and lacking an effective automatic image classification technique it is easiest to manually pre-identify periods featuring pulsating patches.

The luminosity of pulsating patches is frequently described as modulating between 'on' and 'off' states (e.g. Jones et al., 2013; Partamies . In many cases this is an appropriate description but it does not fully describe the behaviours exhibited by the types of pulsating patches outlined in this paper. The different states of patchy aurora can be quite similar, while patchy pulsating aurora does not necessarily disappear during its off state but can instead dim non-uniformly. Furthermore, it may be inappropriate to view the rapid evolution of amorphous pulsating aurora as a transition from one state to another and then back again. In consideration for this, it is more consistent with these classifications to view patches as transitioning between bright and dim states.

Pulsating auroral patches are  typically characterized within keograms by the vertical striations that arise from their pulsations (e.g. Jones et al., 2013; Partamies et al., 2017; Ya . This accurately describes the appearance of amorphous pulsating aurora and – to a lesser extent – patchy pulsating aurora, the two types that pulsate over large portions of their area. However, the appearance of patchy aurora is not well described by this criterion because  its pulsations are generally constrained to small regions and consequently any striations are subtle or absent altogether. Patchy aurora is more easily identified by the pathlines the non-pulsating patches create as they move across the field of view of the ASI. Patchy pulsating aurora (Figure 2b) can exhibit both of these identifying features and are sometimes recognizable in keograms by pathlines that feature vertical striations.

While these observations are true in general, keograms can be insufficient to completely and accurately identify which types of patch are present during a pulsating auroral event. For these situations it is helpful to consult the all-sky images for the period in question. To this end, videos for each keogram in Figures 3 and 4 are included as a supplement (Grono, 2017).

Due to gradation between the types of patches it can occasionally be difficult to categorize a pulsating auroral event. For

5 example, some amorphous pulsating aurora events exhibit more structuring than others and begin to resemble  patchy pulsating aurora. The red-outlined section of Figure 3d between approximately 14:00 and 15:30 UT is one such instance where what appear to be pathlines with striations that would be created by patchy pulsating aurora actually depict what is better described as amorphous pulsating aurora.

**5    Summary and Future Work**

10 In contrast to the results of Yang et al. (2015, 2017), Grono et al. (2017) observed that the motion of certain pulsating auroral patches was inconsistent with convection. This study is an initial step toward identifying the differences between patches that follow convection and those that do not. While many properties of pulsating auroral patches exhibit variation, it is the morphological differences that are highlighted here. In particular, the stability of patches and the spatial extent of their pulsations can be used to categorize pulsating auroral patches. The first criterion describes how rapidly the structure of a patch evolves. A

15 stable patch is one whose shape evolves slowly enough that it can be easily identified throughout a sequence of images, while an unstable patch is difficult or impossible to follow. The second criterion refers to how the region over which a patch pulsates can range from its entire area to small regions near its edge. From these criteria three categories can be defined.

*Patchy aurora* consists of stable patches whose infrequent pulsations are limited to small regions near the edge of larger non-pulsating structures, as seen in Figure 2a. These patches can persist for tens of minutes (Grono et al., 2017) and their

20 motion is consistent with convection.

*Patchy pulsating aurora* includes stable patches that pulsate over most, if not all, of their area. Relative to patchy aurora, these structures are as long-lived and similarly follow convection, however, patchy pulsating aurora have more frequent and prominent pulsations.

*Amorphous pulsating aurora* describes unstable auroral patches that often pulsate over their entire area. In contrast to the

25 patchy auroras, amorphous patches evolve rapidly and can be challenging to identify in subsequent THEMIS all-sky images. The motion of amorphous pulsating aurora is more dynamic than the patchy auroras and may arise due to a different mechanism unrelated to cold, equatorial plasma following convection. Despite these differences  there appears to be a relationship between the mechanisms creating these patches since gradation is observed between the types and it can be difficult to categorize a patch with confidence on occasion.

30 Identifying where the differences between these types originate from is a key question moving forward. Possibilities include the type of plasma wave interacting with the particle precipitation, the source of the plasma wave modulation, and the source region of the particle precipitation. To these ends, the THEMIS spacecraft are well-positioned to provide the necessary data and in a future paper conjunctions between THEMIS ASIs and spacecrafts will be used to investigate these phenomenon.

*Author contributions.* E.G. designed and analyzed the work, and wrote the paper. E.D. is his supervisor and assisted with analysis.

*Acknowledgements.* This research was supported by grants from the Natural Science and Engineering Research Council (NSERC) of Canada and Danish Technical University (DTU). Thanks to Emma Spanswick, Harald Frey, and Stephen Mende for All-Sky data from the NASA Time History of Events and Macroscale Interactions during Substorms (THEMIS) mission.